# OpenReview forum: "Safeguarding Vision-Language Models via Dynamic Token Reweighting"
_ICLR.cc/2026/Conference — ICLR 2026 Conference Withdrawn Submission_

### Official Review · Reviewer_tNBg · 2025-10-15

**Soundness:** 2
**Presentation:** 2
**Contribution:** 2
**Rating:** 2
**Confidence:** 4

**Summary:**

The paper proposed DTR, a multimodal jailbreak defense method that dynamically reweights visual tokens based on Reversal Safety-Relevant Shift (RSS). The paper conducted extensive experiments across several multimodal jailbreak benchmarks. However, the proposed method focuses on VLM jailbreak defense but only considers the visual modality, lacking defenses on textual modality and cross-modal information.

**Strengths:**

1. The method is an inference-time defense and it is easy to implement.
2. The paper conducted extensive experiments across several VLMs and multimodal jailbreak benchmarks.

**Weaknesses:**

1. The method transfers the text-based reweighting defense to the image modality, which lacks novelty.
2. This method only considers the visual modality. As it focuses on VLM jailbreak defense, it lacks considerations on textual and cross-modal information.
3. All experiments are conducted on vanilla datasets, without applying existing attack methods (AutoDAN [1], PAIR [2], UMK [3], BAP Attack[4]) to further optimize on these benchmarks. Therefore, the results cannot genuinely demonstrate the effectiveness of the proposed method.
4. This method seems hard to defend against attacks where the input image is benign but the input text is harmful, like the BAP attack [4]. Although the appendix includes experiments with safe images and harmful text, the effect of DTR is minimal. The ASR-G decreases by only 1.6%, and ASR-R is merely a keyword-based judgement that is not a reliable evaluation metric.
5. The current evaluation only relies on ASR. The authors should include additional metrics, such as using the Perspective API, to provide a more comprehensive assessment.
6. The adaptive attack presented in the paper cannot demonstrate the robustness of DTR. The authors should evaluate DTR against existing strong jailbreak methods as adaptive attacks to validate its robustness.

[1] AutoDAN: Generating Stealthy Jailbreak Prompts on Aligned Large Language Models. ICLR 2024.
[2] Jailbreaking Black Box Large Language Models in Twenty Queries. arXiv:2310.08419.
[3] White-box multimodal jailbreaks against large vision-language models. ACM MM, 2024.
[4] Jailbreak Vision Language Models via Bi-Modal Adversarial Prompt. IEEE TIFS, 2025.

**Questions:**

See Weaknesses.

---

### Official Review · Reviewer_PXaN · 2025-10-26

**Soundness:** 2
**Presentation:** 3
**Contribution:** 1
**Rating:** 2
**Confidence:** 4

**Summary:**

This paper proposes to reweight the visual tokens in a VLM in order to defend against multimodal jailbreak attacks.

**Strengths:**

Paper attempts a novel idea of reweighting the tokens for detecting jailbreak attacks, and defending against them by pushing the representations to be more aligned with the safety-refusal direction

**Weaknesses:**

There are several weaknesses in the work at its current state:
1. The work heavily relies on existence of a single refusal direction in LLMs. The way the authors compute this direction is understandable, but existence of such a vector alludes to the extremely easy separation of the harmful and harmless prompts in the embedding space more than it does to the existence of such a vector when actual harmful prompts will be found in the real world. The datasets they use to compute this vector (advbench and alpacaeval) are so different from each other that the vector has a large magnitude, not to mention their extremely small size just 32 datapoints. If the authors instead used two datasets that are harder to distinguish, they would realize such a vector either doesn't exist or has a small magnitude that it hard to optimize over.

2. Further the optimization objective (Equation 7) tries to weight the visual tokens so that the new representation is aligned towards the refusal direction, while not being different from the original representation (the second term in the equation). These are impossible objectives put together. How can the reweighting the tokens change the representation for safety but also not change them enough from the original representation.

3. Finally the impact of such reweighting on the benign queries is important to understand. The author conduct this experiment on the MM-Vet dataset which is very small, can they conduct this experiment on a much larger dataset. Also can the author confirm that they are using the exact same hyperparameters for the harmful and benign cases in these experiments and the hyperparameters are not tuned to the specific datasets?

4. Can the authors add an average column in Tables 1 and 2 for ease of comparison.

5. In Figure 4, the authors mention existence of large and small alpha values corresponding to the semantic and adversarial tokens -- since the images were not adversarially optimized how did the images even get the adversarial tokens in the first place?

**Questions:**

Please see the weaknesses section.

---

### Official Review · Reviewer_TLoP · 2025-10-30

**Soundness:** 2
**Presentation:** 2
**Contribution:** 3
**Rating:** 4
**Confidence:** 3

**Summary:**

This paper proposes a method of reweighting visual tokens in VLM's that renders jailbreaks unsuccessful. The key insight is that incorporating visual inputs can induce a safety-relevant distributional shift in the model’s activation space, disrupting alignment with the internal refusal direction that governs whether the model rejects or complies with harmful queries. The proposed method DTR seeks to counteracts this induced misalignment by reweighting the visual tokens.

**Strengths:**

## Strengths
- **Novelty** As far as I am aware, very little prior work has studied defending against jailbreaks by reweighting visual tokens - this is quite an interesting direction.
- **Main Results are Compelling** In the main results, DTR appears to significantly reduce ASR and seems to preserve performance on benign inquiries for the most part

**Weaknesses:**

## Weaknesses
- **Data Dependence**   The authors critique existing safeguard measures for relying on curated data (lines 38–40). However, it seems that the effectiveness of DTR is heavily dependent on how accurately the d_ref is estimated — which, in turn, depends on the quality and diversity of the datasets D_harmful and D_harmless. The paper fails to analyze if and why DTR is more sample efficient than standard safety enhancing post-training techniques. Some sensitivity analysis on different choices of D_harmful/harmless (Figure 5 is a good starting point, but does not vary the choice of D_harmful/harmless). What happens if D_harmful/harmless is very different from the adversarial attack chosen at test time?
- **Alternative KV-Cache Eviction Methods** There are many existing KV-Cache methods that exist for the purposes of accelerating inference [1,2]. Do these also enhance robustness to jailbreaks? In Figure 4, the visualization shows that the tokens that are upweighted/retained are the foreground tokens - this is qualitatively similar to how other KV-Cache methods behave. Furthermore, is DTR complementary to existing KV-Cache eviction techniques?
- **Text-only Adversarial Attacks not Discussed** Methods like GCG [3] can also add tokens to the prefix of the prompt that are highly OOD and may also induce a distribution shift that is misaligned with the refusal direction. Is there a reason that the authors focussed on vision only? In practice, a user would likely want to defend against multiple types of attacks (rather than deploying a defense against visual jailbreaks only) so experiments or extensions to text attacks would strengthen the case for this paper.

## References
- [1] An Image is Worth 1/2 Tokens After Layer 2: Plug-and-Play Inference Acceleration for Large Vision-Language Models, ECCV 2024
- [2] DivPrune: Diversity-based Visual Token Pruning for Large Multimodal Models, CVPR 2025
- [3] Universal and Transferable Adversarial Attacks on Aligned Language Models, arxiv

**Questions:**

See weaknesses

---

### Official Review · Reviewer_Lycf · 2025-10-30

**Soundness:** 3
**Presentation:** 3
**Contribution:** 2
**Rating:** 6
**Confidence:** 3

**Summary:**

This paper proposes DTR, an inference-time defense for multimodal jailbreaks that optimizes a VLM’s KV caches to dynamically reweight visual tokens along a learned refusal direction, using a “reversal safety-relevant shift” objective to suppress adversarial visual influence without converting images to text or relying on curated safety references; with early stopping and token eviction for efficiency, DTR significantly lowers attack success rates across HADES, MM-SafetyBench, and JailbreakV-28K while largely preserving benign performance on MM-Vet/MME and adding minimal runtime overhead.

**Strengths:**

1. The writing is clear, and the paper is easy to follow
2. Conduct the comprehensive evaluation on four VLMs and multiple datasets, and demonstrate superior performance on both adversarial and benign scenarios
3. Ablation study supports the effectiveness of the proposed methods

**Weaknesses:**

Main Concerns
1. The proposed method requires multiple forward passes per input to update the alpha values, which may increase inference latency, especially in multi-turn conversation settings.
2. The adaptive attack scenario assumes the attacker only has partial access to alpha. However, the defense could be easily bypassed if the attacker gains full access, which raises concerns about its robustness.
3. What is the performance on the dataset that measures the exaggerated safety aspect of VLMs?
4. What is the default eviction rate in the paper? If we do not do the eviction, what are the ASR in the main table?

Minor Concern
1. The proposed approach resembles prior work on adaptive steering and learning-to-steer methods [1, 2], yet these related papers are not discussed in the related work section.

[1] Steering away from harm: An adaptive approach to defending vision language model against jailbreaks. CVPR 2025

[2] Learning to Steer: Input-dependent Steering for Multimodal LLMs. Arxiv 2508.12815

**Questions:**

1. How many layers does this method need to do the scaling?

---

### Note · Authors · 2025-11-12

I have read and agree with the venue's withdrawal policy on behalf of myself and my co-authors.